# How Can Tufa Deposits Contribute to the Geotourism Offer? The Outcomes from the First UNESCO Global Geopark in Serbia

**Marko D. Petrović** [1,2,*], **Dobrila Lukić** [3], **Milan M. Radovanović** [1,2], **Ivana Blešić** [2,4], **Tamara Gajić** [1,2,5], **Dunja Demirović Bajrami** [1,2], **Julia A. Syromiatnikova** [2], **Đurđa Miljković** [4], **Sanja Kovačić** [4] and **Marija Kostić** [5]

1 Geographical Institute Jovan Cvijić Serbian Academy of Sciences and Arts, 11000 Belgrade, Serbia
2 Institute of Sports, Tourism and Service, South Ural State University, 454080 Chelyabinsk, Russia
3 Eighth Belgrade Grammar School, 71 Grčića Milenka, 11000 Belgrade, Serbia
4 Faculty of Sciences, Department of Geography, Tourism and Hotel Management, University of Novi Sad, 21000 Novi Sad, Serbia
5 Faculty of Hotel Management and Tourism, University of Kragujevac, 36210 Vrnjačka Banja, Serbia
* Correspondence: m.petrovic@gi.sanu.ac.rs

**Abstract:** The study focuses on the present state and the assessments of geotourism development of the two most representative tufa deposits in the Djerdap National Park—the first UNESCO Global Geopark in Serbia. The findings were designated through implementing the freshly upgraded methodology—M-GAM-1-2 based on an early modified geosites assessment model (M-GAM). To overcome the limitations of the previous model, the authors implemented additional enhancements and involved members of the local community (residents and authorities) in the study to comprehensively evaluate the observed sites. The outcomes revealed that the attitudes of all stakeholders should be taken into consideration in order to develop geotourism properly, additionally attract visitors, and preserve tufa deposits for future generations of locals and visitors. Moreover, geotourism at the observed sites can be one of the vital activities of the population, as well as a type of compensation for various limitations in the development, which are imposed by the regimes of natural and cultural heritage protection within the recently established UNESCO Global Geopark.

**Keywords:** Djerdap UNESCO Global Geopark; comprehensive opinions; community involvement; evaluation

## 1. Introduction

Geotourism represents a special market niche of the travel industry motivated by the visits to the geoheritage sites (geosites). A fast development of this aspect of tourism is noticeable in many locations throughout the world; therefore, the literature reported on numerous examples of sustainable development and the significance of geotourism in the areas where it is present [1–12]. Moreover, the development of geotourism should aim to minimize the negative consequences of mass tourism at destinations based on geological and geomorphological attractions. Its main mission is to support sustainable tourism and preserve the environment [9,13]. One of the most comprehensive definitions of geotourism from the environment and economic perspectives has been defined by Dowling [14]. He described geotourism as "sustainable tourism with a primary focus on experiencing the earth's geological features in a way that fosters environmental and cultural understanding, appreciation and conservation, and is locally beneficial. It is about creating a geotourism product that protects geoheritage, helps build communities, communicates and promotes geological heritage and works with a wide range of different people" [14]. In the broader sense, the focus of geotourism relies on nature-based and community-based tourism development with a primary focus on geological features and the landscape [14].



In addition, this reinforces the ABC concept (abiotic-biotic-culture) in geotourism which has been supported by the Arouca Declaration [15]. This document reported geotourism as "tourism which sustains and enhances the identity of a territory, taking into consideration its geology, environment, culture, aesthetics, heritage and the well-being of its residents" (Article 1). Recently, a similar approach has been adopted in several studies [16–19].

Since the focus of geotourism is a visit to the geosite (by visitors—geotourists), this aspect of tourism is directly connected with geoheritage locations and geoparks. In particular, the most common practice of geotourism comprises profiled visits of geological, geomorphological, archeological, mountainous, cultural, and ecological character, which promote the integral value of the area, i.e., tours which have recreational and educational character, in addition to familiarization with geoheritage [20,21]. The base of geotourism development is exclusively the offer of geodiversity, i.e., the geographical diversity of a certain area expressed by the geological structure and morphological elements and processes [22–24]. With the exception of rocks, geomorphological forms, and soil, geodiversity also includes various hydrological and climate processes under the influence of their modification. Phenomena and forms of exceptional significance are singled out from the geodiversity that constitutes geoheritage, which is the reason why they should be protected as natural assets. The fact that geoheritage can teach us the history of the creation of the Earth and the development of the natural world is a very important criterion when determining whether a facility will be placed under protection [25–29].

In this respect, a geopark has an essential role in protecting and promoting geoheritage. According to the UNESCO definition, geoparks are "nationally protected areas with a number of geoheritage sites of particular importance, rarity, or aesthetic appeal" [30]. The foundation of geoparks additionally contributes to non-traditional economic progress based on landscape and geotourism [28,31]. As Ólafsdóttir and Dowling articulated [32], the main purpose of a geopark establishment can be defined as a way of additional pull effects for rural tourism development and an important role in the progress of sustainable local communities. Authors concluded that "both geoparks and geotourism may be seen as attractive endeavors for rural development in many peripheral areas facing emigration" [32].

Among the sites in geoparks, karst landforms represent an important component in the international movements of geotourists. Karst areas in Serbia comprise several geotourists destinations recognized on the tourism market, among which Djerdap UNESCO Global Geopark stands out. The geosites most visited by tourists are canyons, gorges, karst springs, and waterfalls [27]. However, the geosites with an international status, such as IUGS Global Geosites, World Heritage Sites, and UNESCO Global Geoparks, have a leading role in the advancement of this form of tourism [28]. Additionally, morphohydrological complexes of karst springs and tufa deposits (tufa—calcareous sinter) represent a special sphere of the offer which has still not been sufficiently presented in the scientific literature, despite the clear high demand for the visits to these locations [33–35]. Moreover, if they are within the protected natural assets with an international status, such as UNESCO Global Geopark, the possibility of their promotion is higher, and the offer for the visitors is more diverse and increasingly attractive.

The importance of tufa deposits is completely minimized and neglected in the scientific literature on tourism in comparison with the recognition of other geosites on the newly established Djerdap UNESCO Global Geopark [36–38] and other parts of Serbia [39–43]. In this respect, the study aims to quantively analyze the most relevant tufa deposit sites in the geopark and assess their significance for the national and global geotourism progress. The paper's principal aim is to investigate the applicability of each presented subindicator for the estimation procedure by submitting the importance factor in the newly introduced method. Moreover, the objective of the paper will be directed toward the assessment of the significance of the observed sites in the tourism of the first UNESCO Global Geopark in Serbia. The major goal of the study is to demonstrate this objective by combining the established and upgraded geosite assessment model (GAM) to the chosen locations, in

which the subindicator scores show preference to one geosite over another. Thereafter, the outcomes of the examination will show how the distinctions in importance for each individual subindicator might impact the final result. The conducted valorization would additionally contribute to the rural development of Eastern Serbia, as one of the less advantaged parts of the country. In this respect, a comprehensive model for the assessment of the observed sites will be proposed for the evaluation of the present condition and guidelines for other geosites. The two selected representative locations are quantitatively assessed by applying the modified geosite assessment model (M-GAM). In addition to the frequent assessment of the opinions of experts and visitors in the M-GAM evaluation to date [40,44], the model will be upgraded by the attitudes of the local community (M-GAM-1) and local authorities (M-GAM-2). In this way, the integrated perception of the significance and role of the geosites will be created in the academia, science, society, and local community. This is the reason why the existing policies for the enhancement of geotourism, along with the founding of the environmental development conception, mostly refer to the similar geosites in the country and abroad. Selected sites represent the geosites of exceptional scientific, educational, and tourism significance, which was the reason why they were selected for this analysis. The basic criterion for selecting them was previous field trips and studious research conducted in this part of Serbia as well as knowledge of conditions at the field, and the theoretical and empirical background of the selected geographical area. The geological representative features of the chosen locations are unique on the national level [36], and there are no similar sites within Djerdap UNESCO Global Geopark that could be compared to them.

The present section represents the introducing remarks including a short theoretical background on geotourism and geosites. The following sections will shed more light on the study area and selected geosites, provide implementations of the modified model revealing the original findings, and finally present the discussion and conclusions in detail.

## 2. Study Area and Overview of the Selected Sites

The Djerdap National Park and its wider surroundings, with the total area of 1330 km$^2$, was declared as the first geopark in Serbia in 2020. The official name of this protected area is Djerdap UNESCO Global Geopark, which made it a part of the Global Geopark Network [45]. In this respect, the exceptional international values of the wider area of Djerdap were recognized since its features are highly compatible with the nature protection role and they have great significance for the research of the development of the Earth's crust [46–49]. The mentioned key values of geodiversity are accompanied by diversified biodiversity [50,51], as well as cultural, historical, ethnological, archeological, and tourism sites. The main purpose and objective of forming Djerdap UNESCO Global Geopark are the recognition, protection, preservation, presentation, promotion, management, and responsible consumption of available resources, along with the education of local communities with the aim of sustainable development. This was reported and highlighted in many recent studies [6,27,36–38].

Djerdap UNESCO Global Geopark is located in Eastern Serbia, in the border area with Romania, and it is a part of the Carpathian Mountains, stretching in the west-east direction. This mountainous area has been formed in quite diverse geological conditions, including the intensive rise, breaking, modification of secondary erosion, and intensive processes of karstification. The outline of Djerdap Geopark consists of the monumental Djerdap Gorge (Figure 1), a composite, polyphase, and antecedental valley of the Danube River, as well as of the places with exceptional characteristics of geological history. This primarily refers to the stratigraphic and paleontological profiles and sites from the Jurassic and Cretaceous periods with the abundance of fossils, forms of karst and fluvial-karst landforms, primarily caves and pit caves, of great length and depth, of interesting morphology and morphogenesis, morpho-hydrological complexes of karst springs, tufa deposits and karts bridge [52].

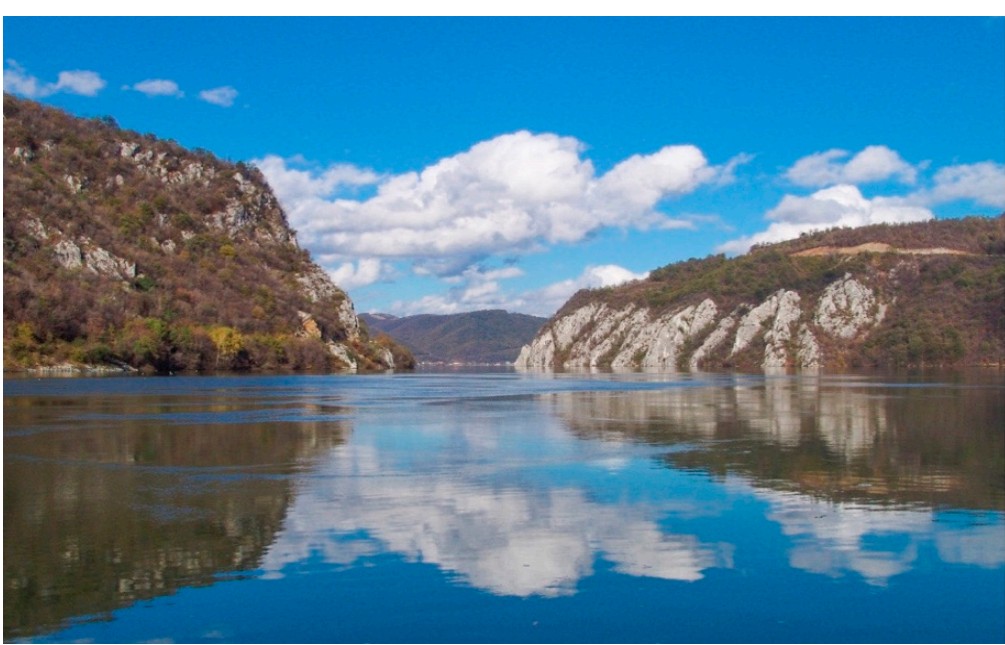

**Figure 1.** The Iron Gate (Djerdap Gorge) on the Danube River—the main attraction of the geopark (Photo: D. Lukić, 2021).

In the area of Djerdap UNESCO Global Geopark, 63 different sites of geoheritage were identified which have international protection [45] within the programs of Emerald Network (since 2007), Ramsar Convention (since 2020), IPA Programme (since 2006), IBA Programme (since 2020), and PBA Programme (since 2008). Furthermore, this area has been placed on the Preliminary UNESCO list (since 2002), on the list of Carpathian areas (Carpathian Convention, BioREGIO Carpathians project, etc.), ICPDR (since 2003), the DanubeParks (since 2009), and is planned for inclusion in the UNESCO—Man and the Biosphere (MAB) Programme.

Among the protected sites, the prevailing ones are karst landforms, and those which stand out are the sites of the tufa deposits near Tumane (or Tuman) Monastery (G1) and Beli izvorac tufa deposits (G2) (Figure 2). Both sites, due to their morphological features (tufa, waterfalls, springs, caves, etc.) and well-preserved state, represent some of the most significant and outstanding locations of the karst landscape in the country [53], which is the reason why they are placed on the national list of the relevant geosites as natural monuments and selected for evaluation in this study. In contrast to other rocks formation, tufa can be rapidly transformed due to natural and anthropogenic disturbances. In addition, the vulnerability of this natural phenomenon is pronounced, and protection is highly required [54]. Their profiles have been presented in Figure 3a,b.

G1: Tufa deposits near Tumane Monastery are situated in the Municipality of Golubac. The deposits are situated in the basin of the Tumanska River, on the right valley bank of the Kamenica stream and the western slopes of the Severni Kučaj Mountain. The stream valley is formed on the fault of the meridian direction of movement, along which Paleozoic schist formations occurred over the Jurassic and Cretaceous limestone. This is how the schist hills Crveni kamen (406 m) and Crni vrh (591 m) were formed on the western side of the valley, as well as Tilva (561 m), the hill of Jurassic limestones, on the eastern side of the valley. At the bottom of the right valley side of the Kamenica stream, in the central part of the valley, at an altitude of 250 m and 1.1 km from its flow into the Tumanska River [52,55], tufa deposits have been formed (Figure 3a).

G1 resembles a terrace of a fanlike shape, with a relative height of 14 m, and covers an area of 8550 m$^2$ (Figure 4). It was formed on the place where a periodical spring outflows at the contact of limestones and schists at an altitude of 268 and 18 m above the Kamenica stream bed [55]. Moreover, this spring is conditioned by the existence of the fault

which transversely cuts the stream valley a bit downstream from the tufa deposits and separates the limestones from the schists on the north. Furthermore, it is exclusively fed by the precipitation that falls on the Tilva hill. Water is always present on the faucet, but with varying discharge. Tufa depositing has stopped and the process of formation of this vulnerable rock is not active anymore, which is the result of the degradation by the water intake structure. If it were moved below the deposit formation, tufa would accumulate again, and the whole ambient unit would recover its previous state and become even more important as a site of geoheritage at the national level. Additionally, it would be even more attractive for tourists' visits. Low discharge of the spring points to the fact that the tufa deposits were formed in the period of a significantly humid climate, most probably during the Atlantic phase of the Holocene [52]. After the formation of the tufa deposits near Tumane Monastery, the Kamenica stream bed was accumulated with rock material of several meters, whereas on the tufa deposits, a soil cover of 1 m in thickness was formed and a thick beech forest still exists [56].

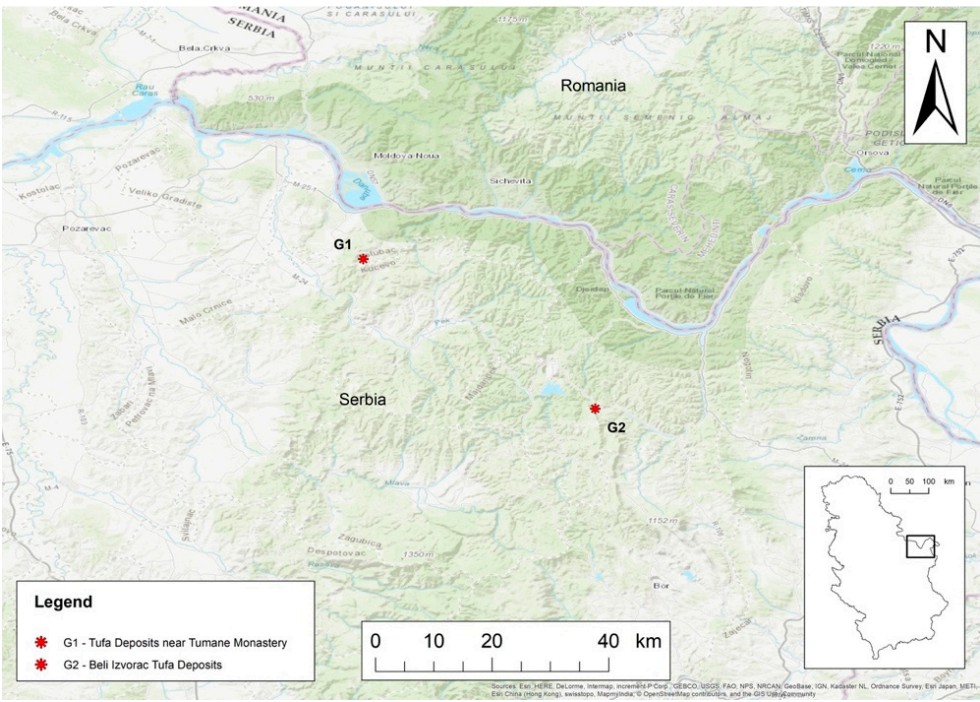

**Figure 2.** The positions of the G1-G2 sites in Eastern Serbia (D. Lukić, 2022).

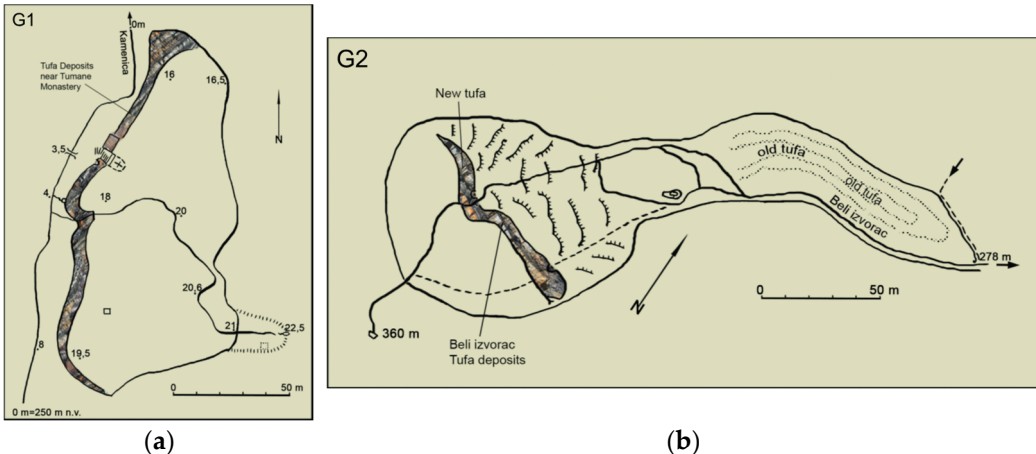

**Figure 3.** (**a,b**) Tufa deposits profile sketch (indicated by shaded lines) on the G1 and G2 sites (Adapted from Gavrilović [52]).

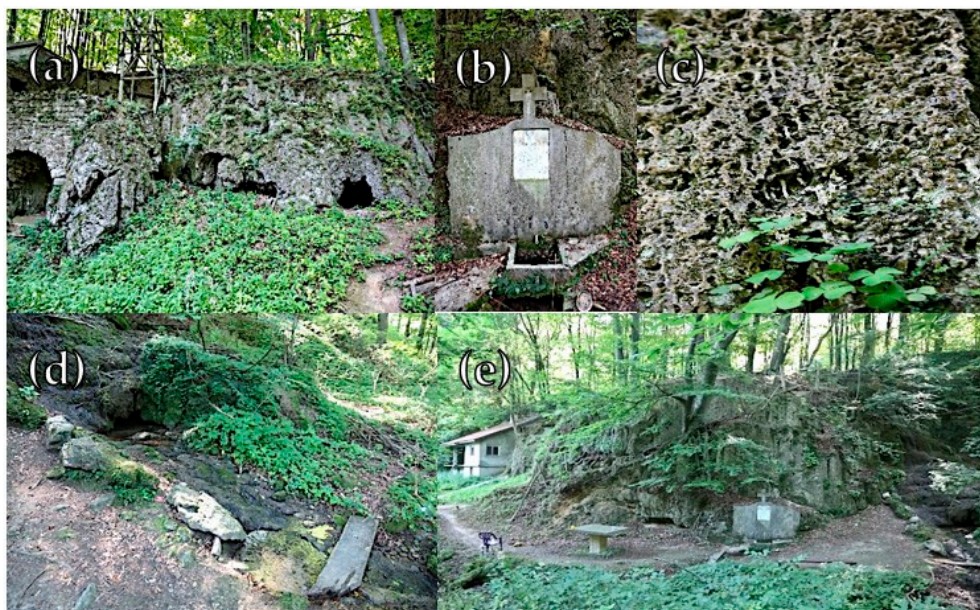

**Figure 4.** Photos of the presented G1 site. (**a**) The front main profile of the tufa deposits; (**b**) the fountain at the foot of the tufa deposits; (**c**) the tufa of the streamlet that flows into the Kamenica stream; (**d**) the spring above the bed of the Kamenica stream; (**e**) the area on the right bank of the Kamenica stream with the fountain on the slightly right side and hermit-church on the far left side (Photos: D. Lukić, M.D. Petrović, M.M. Radovanović, 2019).

Previous research of the G1 site has been primarily focused on its geomorphological and hydrological characteristics [52,55,56]. On this site, a great anthropogenic impact is noticeable, bearing in mind that in the 1950s, on the tufa, hermit Pahomije built a hermit-church 5 × 4 m in size which was unique in Serbia, next to a living area with a belltower where the hermit lived. The tufa deposit with the hermit-church and picturesque surrounding is a place which is often visited by various visitors, both domestic and foreign (especially after the rapid development of religious tourism in the nearby Orthodox Tumane Monastery), school and research excursions, but there is still no proper information about this geosite for the wider public. This was one of the reasons why this site was selected for further evaluation and valorization.

G2: Beli izvorac tufa deposits are situated in the basin of the Šaška River, in the Municipality of Majdanpek. The site is located on the north slopes of the Krš mountain range (it includes the Goli krš, the Stol, the Veliki krš, and the Mali krš Mountains), which was mostly formed of Jurassic and Cretaceous limestones. This area is rich in karst landforms, presented by sinkholes, uvalas, blind valleys, and caves. The cave from which the Beli izvorac spring flows is located in the massive Upper Jurassic limestones (Figure 5). Magma rocks, granodiorite, from the Old Paleozoic lean on those limestones in the east, and in the west, on Cretaceous pyroxene-andesites [57].

The Beli izvorac stream (1250 m in length) flows under the Krš cut from a cave at 360 m a.s.l. (Figure 3b) and drains into the Šaška River at 230 m a.s.l. Under the limestone section from the north toward the south, three caves appear, with all of them approximately the same height: Mala pećina cave, Kozja pećina cave, and Beli izvorac cave, and they represent the phases of the lowering of the underground flow [58]. In its central part, the stream valley is narrow and approximately 150 m in depth, with an area of drainage basin of 0.8 km². The total area of tufa deposits is 12,000 m² [55].

In the cave canal, where the Beli izvorac spring flows, there is a thick tufa accumulation. After it flows out from the cave at the length of 30 m, no formations of calcium carbonate are visible, only to appear again on the wide tufa terrace. The terrace covers an area of 3000 m², and from the lower side it ends with a section of 20 m in height. On the tufa terrace, the Beli izvorac stream has formed the stream bed at the depth of 2–3 m, which

ends with a waterfall of 16 m in height. Under the waterfall, a cave canal was formed, open from both sides and 13 m in length and 4 m in width [59]. The Beli izvorac karst spring has a changeable discharge which varies from 10 to 100 L/s, the highest in spring and autumn, and the lowest during summer and winter. It is fed by the water that originates from a karst aquifer, which is located at the bed rock made of schists and magmatic rocks, the feeding area of which is estimated to be 5 km$^2$ [60]. What is characteristic of this aquifer is that its water remains in the karst underground for a long period of time, and its slow exchange is one of the most important factors of tufa formation in this place. The greatest tufa depositing occurs due to the release of carbon dioxide in the place under the waterfall and in the place of numerous cascades covered with moss that perform the assimilation of this gas for its physiological needs [61].

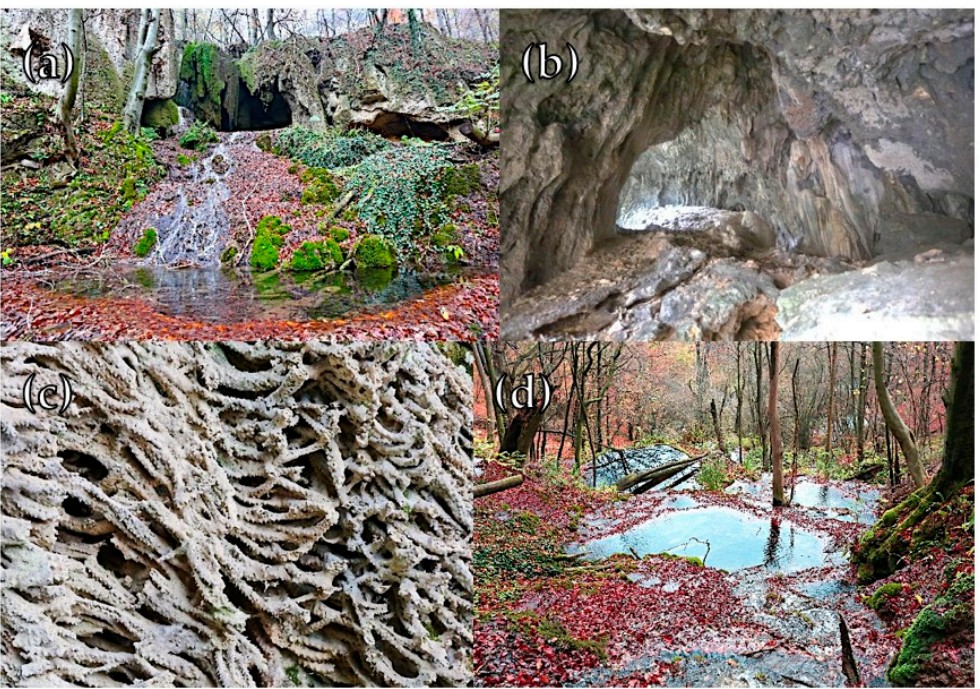

**Figure 5.** Photos of the presented G2 site. (**a**) The main entrance to the cave and the source of the Beli izvorac stream; (**b**) a cave channel under the Beli izvorac waterfall formed by plunging water through the tufa; (**c**) a closer look at the tufa of Beli izvorac; (**d**) the tufa partitions with ponds in the bed of the Beli izvorac stream (Photos: D. Lukić, M.D. Petrović, T. Gajić, 2021).

Previous research of G2 site and its surroundings was based exclusively on hydrological, geomorphological, and geological features [52,55,57–61]. Of note, however, the anthropogenic impact at G2 has been negligible, although there have been tendencies to build water intakes at the area of the Beli izvorac spring, which would endanger the process of formation of tufa deposits, the existence of the attractive waterfall, as well as its flora and fauna. For the abovementioned reasons, this geosite is declared as a protected natural monument [53] and will be examined in this paper.

## 3. Materials and Methods

In this study, the method relies on the developed M-GAM model suggested by Tomić and Božić [44] and tested by a number of authors [8,40,41,62]. M-GAM embodies the reformation of the primary GAM model (Geosite Assessment Model) which was previously proposed by Vujičić et al. [63] and successfully applied by Petrović et al. [64] and Višnić et al. [65]. Moreover, this study aimed to upgrade and compare the existing models in order to achieve a more accurate outcome. In this respect, the suggested M-GAM-1 model includes the assessment of the opinions of the local community in the surroundings of the site of geoheritage. The proposed upgrade of the model is established on the suggestion from an earlier research [66,67],

which points out that the local population should have a significant part in the creation of the geotourism product. Moreover, the newly suggested M-GAM-2 model includes the role of the local government/authorities. This addition to the existing models is justified by the findings of several modern studies [68–71], which have proved a great importance of the local authorities in the protection and promotion of geoheritage sites.

The examination deliberated to compare the results by involving an upgraded M-GAM-1-2 model, and for the first time, introducing the attitudes of the local community (both ordinary people and authorities in the local environment) in the previous, starting point model. Thanks to the suggested additions, it is possible to holistically perceive the role that the observed geoheritage sites have in the society and in the tourism of the observed area. The development of the model is shown in Figure 6.

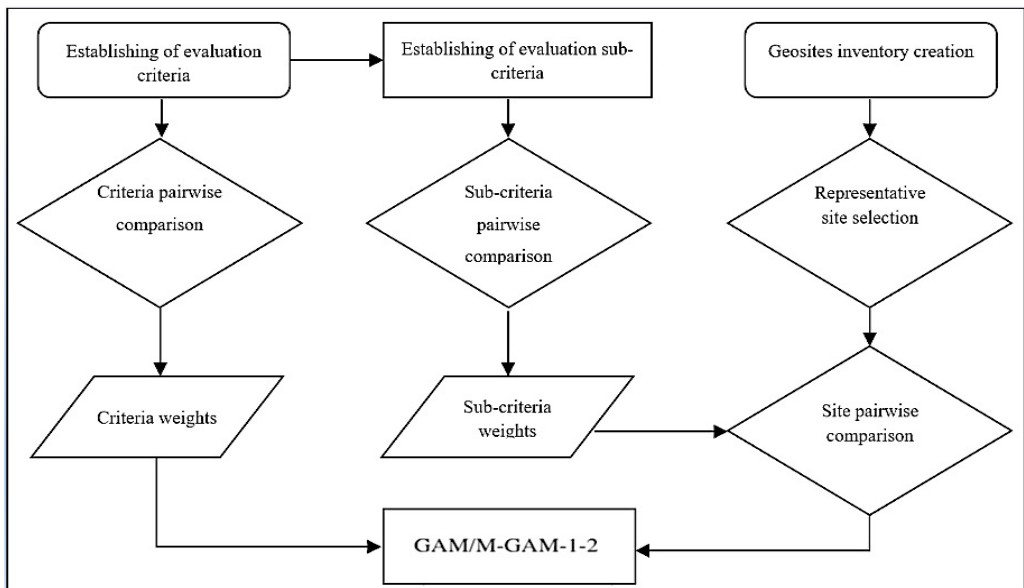

**Figure 6.** The method flowchart.

The original GAM model analyzes main values (MV) with 12 subindicators and additional values (AV) with 15 subindicators, as a basic starting point (Figure 7). The exact scores and description of subindicators are adapted from Petrović et al. [8] (p. 447). The sampling technique was a questionnaire based on the original GAM subindicators focusing exclusively on experts' opinions. The same parameters will be updated for the modified versions M-GAM, M-GAM-1, and M-GAM-2, which will contribute to the new perception and grading of the observed sites and upgrade the existing model. This will involve not only experts' and visitors' opinions but the attitudes of the surrounding local people and municipal authorities for the very first time. In addition, it will be the first grading of this kind since the declaration of Djerdap UNESCO Global Geopark by the General Conference of UNESCO in 2020.

The research also included the importance factor ($1 \geq Im \geq 0$) [44], which provides the respondents with a chance to clearly show the attitude toward any statement in the scale. It was necessary to include the local community mostly since visitors and experts can assess certain professional and market aspects of geosites, but not those that refer to the improvement of the local community. Moreover, visitors and experts performed their evaluations from the travel or scientific perspective, which are, as research shows, less important for the local benefits [68,71]. Nevertheless, we anticipate that the opinions of all the observed stakeholders should provide more objective and holistic results. The local community can assess the subindicators in the same way as visitors and experts for the MV

and AV. Both values represent the sum of subindicators shown in Figure 8 which make up the original GAM model:

$$GAM = MV\ (VSE + VSA + VPr) + AV\ (VFn + VTr). \tag{1}$$

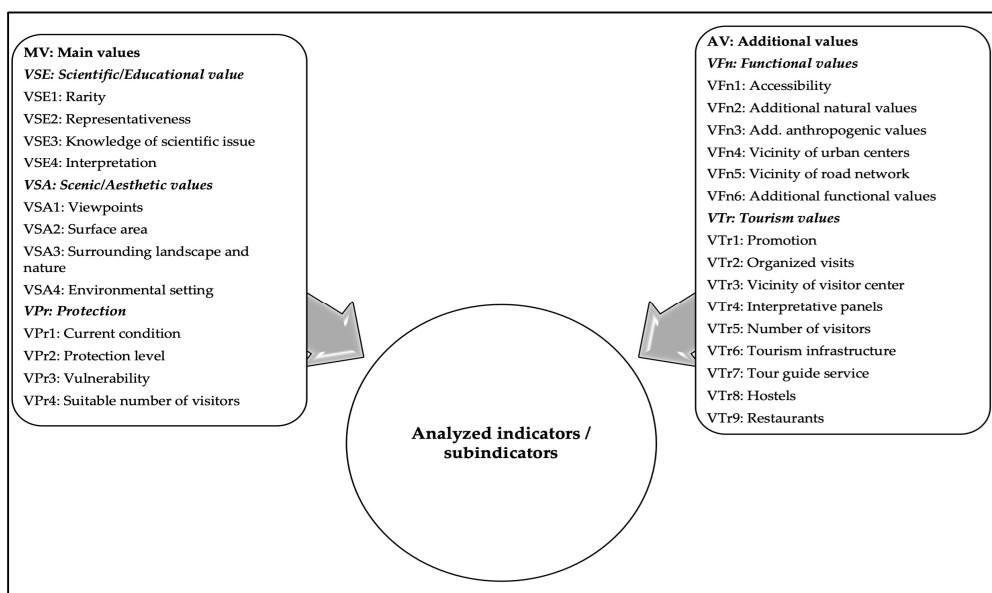

**Figure 7.** Structure of the primary model (Adapted from Vujičić et al. [63]).

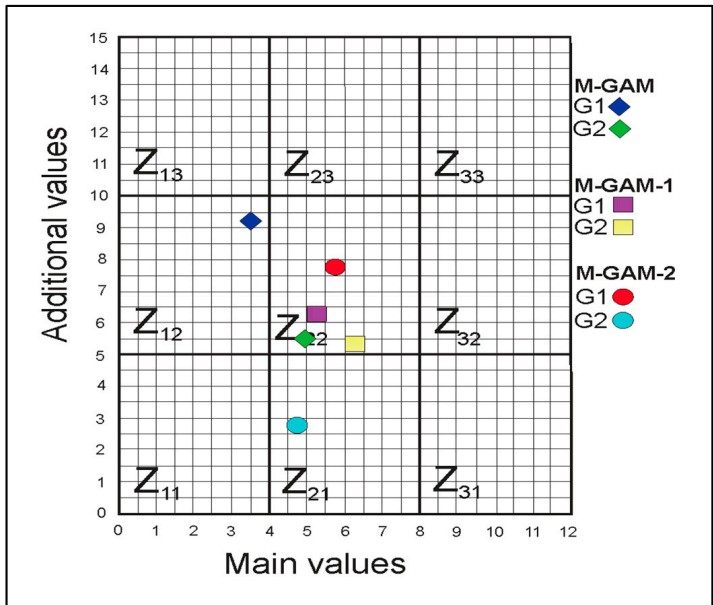

**Figure 8.** Positions of G1-G2 sites in M-GAM-1-2 matrix.

Following the equation, M-GAM-1-2 models also include *Im* in the existing model:

$$M\text{-}GAM\text{-}1\text{-}2 = Im(GAM) = (MV + AV). \tag{2}$$

In order to achieve *Im*, *K* represents a number of visitors, *Ivk* is a score of the visitors' grades for each subindicator, *K* is the total number of visitors, while *Im* parameter contains any scores from 0.00 to 1.00. Finally, by adding all the values, the following equation is obtained:

$$1\ \geq\ Im \geq 0 = \frac{\sum_{k=1}^{K} Ivk}{K}. \tag{3}$$

The equation indicates that the *Im* is multiplied by the scores specified by the members of the local population and authorities. Therefore, a more realistic assessment is performed as compared to the application of the upgraded earlier model. In particular, the local population and government assess the importance of a subindicator with a score of 0.50, where the final score cannot be 0.75 or 1.00, but it should be lower (0.50) if their opinions are taken into consideration, compared to the opinions of experts and visitors. In this respect, M-GAM-1-2 models show the status of the MV and AV geosites that do not successfully reach their full affirmation. It sheds more light on the potential tourism progress in Djerdap UNESCO Global Geopark, which should be encouraged by the development of the standards and resources which have not been realized yet, but are important for the local area [53].

The research was conducted in the observed sites, surrounding settlements, and local self-governments from September 2019 to August 2022. The sampling was voluntary and anonymous, and the research subjects were the visitors to the site, experts (geographers, geologists, geomorphologists, and tourism managers), the local population (from the settlements surrounding the sites in a radius of approximately 15 km), and the members of the local governments (municipalities and rural community offices) who express the goodwill to present the opinions toward the examined subject. The instrument applied to this study involved 27 statements (subindicators). The questionnaire was properly completed by 366 visitors and 13 experts (M-GAM) at the first stage, and by 55 locals (M-GAM-1) and 11 members of the local government (M-GAM-2) at the second stage. It was divided into two segments: The first segment contained the statements that refer to the MV, whereas the second contained the AV. The mandatory request for each examinee was to assess the *Im* of each subindicator by giving a score that ranges from 0 (not at all important) to 1.00 (very important) on a 5-point Likert scale.

## 4. Results

The findings were obtained by adding the MV with AV and their mean values. The aggregated grades, obtained from all examined stakeholders for each location are shown in Table 1.

In the following sections, the results of Table 2 and Figure 7 will shed more light on the evaluation of main values (MV) and additional values (AV). Table 2 illustrates MV = 5.07 and AV = 6.14 as the average values in M-GAM-1-2 model.

**Table 1.** Estimation of examined stakeholders for G1-G2 sites (0.00–1.00).

| *GAM* | Complete Scores | | | | | | *Im* | Scores by Experts | | Scores by Locals | | Scores by Authorities | |
|---|---|---|---|---|---|---|---|---|---|---|---|---|---|
| | G1 | G2 | G1 | G2 | G1 | G2 | | G1 | G2 | G1 | G2 | G1 | G2 |
| **Scientific/educational values** | | | | | | | | | | | | | |
| *VSE1* | 0.22 | 0.65 | 0.43 | 0.86 | 0.86 | 0.65 | *0.85* | 0.25 | 0.75 | 0.50 | 1.00 | 1.00 | 0.25 |
| *VSE2* | 0.60 | 0.20 | 0.40 | 0.80 | 0.40 | 0.40 | *0.74* | 0.75 | 0.25 | 0.50 | 1.00 | 0.50 | 0.50 |
| *VSE3* | 0.61 | 0.41 | 0.61 | 0.81 | 0.41 | 0.20 | *0.75* | 0.75 | 0.50 | 0.75 | 1.00 | 0.50 | 0.25 |
| *VSE4* | 0.17 | 0.51 | 0.68 | 0.68 | 0.68 | 0.68 | *0.68* | 0.25 | 0.75 | 1.00 | 1.00 | 1.00 | 1.00 |
| **Scenic/aesthetic values** | | | | | | | | | | | | | |
| *VSA1* | 0.23 | 0.25 | 0.50 | 0.50 | 0.25 | 0.50 | *1.00* | 0.23 | 0.25 | 0.50 | 0.50 | 0.25 | 0.50 |
| *VSA2* | 0.23 | 0.46 | 0.23 | 0.23 | 0.46 | 0.23 | *0.54* | 0.25 | 0.50 | 0.25 | 0.25 | 0.50 | 0.25 |
| *VSA3* | 0.14 | 0.14 | 0.57 | 0.43 | 0.43 | 0.14 | *0.75* | 0.25 | 0.25 | 1.00 | 0.75 | 0.75 | 0.25 |
| *VSA4* | 0.14 | 0.14 | 0.14 | 0.14 | 0.57 | 0.14 | *0.92* | 0.25 | 0.25 | 0.25 | 0.25 | 1.00 | 0.25 |
| **Protection values** | | | | | | | | | | | | | |
| *VPr1* | 0.50 | 1.00 | 0.50 | 0.50 | 0.25 | 0.50 | *1.00* | 0.50 | 1.00 | 0.50 | 0.50 | 0.25 | 0.50 |
| *VPr2* | 0.27 | 0.54 | 0.54 | 0.54 | 0.27 | 0.54 | *0.92* | 0.50 | 1.00 | 1.00 | 1.00 | 0.50 | 1.00 |
| *VPr3* | 0.19 | 0.19 | 0.37 | 0.56 | 0.37 | 0.37 | *0.57* | 0.25 | 0.25 | 0.50 | 0.75 | 0.50 | 0.50 |
| *VPr4* | 0.23 | 0.23 | 0.46 | 0.46 | 0.46 | 0.46 | *0.57* | 0.25 | 0.25 | 0.50 | 0.50 | 0.50 | 0.50 |

**Table 1.** *Cont.*

| GAM | Complete Scores | | | | | | Im | Scores by Experts | | Scores by Locals | | Scores by Authorities | |
|---|---|---|---|---|---|---|---|---|---|---|---|---|---|
| | G1 | G2 | G1 | G2 | G1 | G2 | | G1 | G2 | G1 | G2 | G1 | G2 |
| **Functional values** | | | | | | | | | | | | | |
| VFn1 | 0.43 | 0.85 | 0.85 | 0.85 | 0.64 | 0.21 | *0.50* | 0.50 | 1.00 | 1.00 | 1.00 | 0.75 | 0.25 |
| VFn2 | 0.74 | 0.74 | 0.74 | 0.37 | 0.74 | 0.19 | *0.77* | 1.00 | 1.00 | 0.50 | 0.50 | 1.00 | 0.25 |
| VFn3 | 0.75 | 0.56 | 0.75 | 0.75 | 0.75 | 0.19 | *0.93* | 1.00 | 0.75 | 1.00 | 1.00 | 1.00 | 0.25 |
| VFn4 | 0.85 | 0.43 | 0.85 | 0.85 | 0.85 | 0.21 | *1.00* | 1.00 | 0.50 | 1.00 | 1.00 | 1.00 | 0.25 |
| VFn5 | 1.00 | 0.25 | 0.25 | 0.25 | 1.00 | 0.25 | *1.00* | 1.00 | 0.25 | 0.25 | 0.25 | 1.00 | 0.25 |
| VFn6 | 0.46 | 0.68 | 0.46 | 0.46 | 0.91 | 0.23 | *0.91* | 0.50 | 0.75 | 0.50 | 0.50 | 1.00 | 0.25 |
| **Tourism values** | | | | | | | | | | | | | |
| VTr1 | 0.13 | 0.00 | 0.13 | 0.00 | 0.25 | 0.00 | *0.75* | 0.25 | 0.00 | 0.25 | 0.00 | 0.50 | 0.00 |
| VTr2 | 0.77 | 0.19 | 0.19 | 0.19 | 0.19 | 0.00 | *0.74* | 1.00 | 0.25 | 0.25 | 0.25 | 0.25 | 0.00 |
| VTr3 | 0.93 | 0.23 | 0.23 | 0.23 | 0.23 | 0.23 | *0.85* | 1.00 | 0.25 | 0.25 | 0.25 | 0.25 | 0.25 |
| VTr4 | 0.50 | 0.25 | 0.25 | 0.25 | 0.25 | 0.00 | *0.85* | 0.50 | 0.25 | 0.25 | 0.25 | 0.25 | 0.00 |
| VTr5 | 0.86 | 0.22 | 0.22 | 0.22 | 0.22 | 0.22 | *0.86* | 1.00 | 0.25 | 0.25 | 0.25 | 0.25 | 0.25 |
| VTr6 | 0.25 | 0.25 | 0.25 | 0.25 | 1.00 | 0.25 | *1.00* | 0.25 | 0.25 | 0.25 | 0.25 | 1.00 | 0.25 |
| VTr7 | 0.25 | 0.25 | 0.25 | 0.25 | 0.25 | 0.25 | *1.00* | 0.25 | 0.25 | 0.25 | 0.25 | 0.25 | 0.25 |
| VTr8 | 0.50 | 0.25 | 0.25 | 0.25 | 0.25 | 0.25 | *1.00* | 0.50 | 0.25 | 0.25 | 0.25 | 0.25 | 0.25 |
| VTr9 | 0.65 | 0.44 | 0.44 | 0.22 | 0.44 | 0.22 | *0.87* | 0.75 | 0.50 | 0.50 | 0.25 | 0.50 | 0.25 |

**Table 2.** M-GAM-1-2 values scores.

| M-GAM-1-2 | G1/G2 Label | MV | | AV | | Z |
|---|---|---|---|---|---|---|
| | | VSE + VSA + VPr | Σ | VFn + VTr | Σ | |
| **M-GAM** | G1 | 1.60 + 0.76 + 1.19 | 3.55 | 4.23 + 4.84 | **9.07** | Z12 |
| | G2 | 1.77 + 1.24 + 1.96 | 4.97 | 3.51 + 2.08 | **5.59** | Z22 |
| **M-GAM-1** | G1 | 2.12 + 1.19 + 1.87 | 5.18 | 3.90 + 2.21 | 6.11 | Z22 |
| | G2 | 3.15 + 1.05 + 2.06 | **6.26** | 3.53 + 1.86 | 5.39 | Z22 |
| **M-GAM-2** | G1 | 2.35 + 1.96 + 1.35 | **5.66** | 4.89 + 3.08 | 7.97 | Z22 |
| | G2 | 1.93 + 1.01 + 1.87 | 4.81 | 1.28 + 1.42 | 2.70 | Z21 |
| *Mean* | | | *5.07* | | *6.14* | |

Figure 8 represents the M-GAM-1-2 matrices divided into nine fields ($Z_{11}$–$Z_{33}$). The matrix shows the differences between the observed geosites and points to different assessment results obtained from the experts, contrary to the ones from the local residents and members of local authorities. The results of the main values and additional values for both sites are illustrated by the X and Y axes. Presented nine Z fields (i, j = 1, 2, 3) explained the different evaluations they obtained through the assessment process [8].

## 5. Discussion

The results presented in Table 1 indicate that experts scored the G2 site (Beli izvorac tufa deposits) with the highest grades from the perspective of science and education. Their opinions are supported by the fact that G2 is the site with exceptionally rich karst landforms. This is especially important for the research work of geomorphologists, geologists, and biologists, considering the fact that the cave from which the Beli izvorac springs is located in the representative Malmo Upper Jurassic limestones [59] is highly significant for the wider scientific community and educational purposes with a (local) population. Faults formation of rock masses is the result of neotectonic movements that took place during the periods of Pliocene and Quaternary, which is the reason why G2, to date, is one of the scientifically relevant examples of karst profiles in the Carpathians.

Nonetheless, the opinions of the local government gave a higher score to the G1 site (tufa deposits near Tumane Monastery) in comparison with G2. The score was based on their perception of the G1 site as one of the drivers of the development of their local municipality (Golubac) since the site records an extraordinary number of visits made by researchers and organized school excursions [53], which combine the tour at the site with a visit to the widely known Tumane Monastery (one of the most important pilgrimage centers on the Balkans).

The G1 site is primarily recognizable for the geological profile where the formation of layers of Paleozoic shales over the Jurassic and Cretaceous limestones can be noticed, which gives it the status of one of the best-known karst sites in Serbia and the lower Danube Basin. It is necessary to emphasize that G1 and G2 were given high scores since tufa deposits in this part of the Balkans appear as unique phenomena and their forms are rare and highly important for science and education. Although G1 and G2 are not significantly presented in popular scientific publications and media as they should be, these geoheritage sites have great educational and didactical roles. Additionally, they are present in regular geosciences textbooks for elementary/middle schools, high schools, and colleges in Serbia and the surrounding countries as examples of specific forms of karst morphology and hydrography in the Carpathians.

The knowledge of geoscientific issues related to these forms of geoheritage is at a high level taking into consideration that they were the subject of research of eminent experts in geomorphology and hydrology of karst from the most imperative research and academic associations in the country [72,73]. The level of interpretation of tufa deposits as sites of geoheritage could be more simple and interesting bearing in mind that they are very interesting, rare, and unique phenomena of karst morphology and hydrography.

Regarding the scenic and aesthetic values, the experts' opinions differ from the opinions of the local community and government. Namely, while experts assessed G2 in this category as the most significant, the other two groups gave a higher score to the G1 site as more representative and aesthetically more attractive. The results can be explained by a great anthropogenic impact in G1 taking into consideration that the small church of hermit Pahomije was built within that profile as unique religious site, and the absolute majority of residents and members of the local government consider it an additional (complementary, spiritual, aesthetic) value of the site. Moreover, a small water intake was built, which disturbed the aesthetic value of the place, and the deposit itself was dead from the anthropogenic impact. If the water intake was moved out of the tufa deposit G1, it would be reactivated, and the consequences would be immensely positive.

On the other hand, the experts gave higher scores to G2 due to the nature preservation, aesthetics of the surrounding area, and more diverse geodiversity. The environmental settings of G1 and G2 sites, as special aesthetic values in terms of the preservation of natural landscape and good environmental conditions, are at a satisfactory level. However, what is noticeable is the non-existence of viewpoints and surfaces for tourism activities on both sites, which diminishes special aesthetic values [74].

When observing protection values, the total scores were synchronized in all the groups of respondents since G2 received clearly higher scores. Although both sites are placed on the national list of the geoheritage as natural monuments and are under international protection within Djerdap UNESCO Global Geopark, G2 is characterized by an extremely low level of natural or anthropogenic endangerment [59], which was sufficient to receive the highest grades. This can be explained by its relatively isolated position in relation to the busy tourist routes in the geopark. On the other hand, the number of visitors to G1 is significantly higher, with over a million visitors in 2020 [53], and the consequence of which is a higher level of caution of tourist organizations regarding the protection of the site.

Regarding the importance of functional values subindicators, G1 received the highest scores. The observed stakeholders marked all subindicators as the highest. This reinforces the fact that this geosite is situated in the vicinity of the busy Djerdap highway which follows the flow of the Danube (the international water Corridor 7), with numerous cultural



and natural sights in its closest surrounding (Tumane Monastery, Djerdap Gorge, Golubac medieval Fortress, Silver Lake summer resort, Viminacium Roman archeological site, Ram medieval Fortress, etc.). All these sites additionally attract the highest number of domestic and international visitors in the area, especially throughout the last decade [53].

On the other hand, G2 is located in a more isolated part of Eastern Serbia, away from busy roads and well-known sights which could influence higher scores. However, additional functional values of both sites are not at a very high level. The exception is a big parking space next to Tumane Monastery (near G1), which is designed for a large number of cars and travel buses.

The tourism values outcomes revealed that all the observed stakeholders have a favorable opinion toward G1, which has reached the most beneficial scores. Additionally, the accommodation and restaurant services in the vicinity of G1 are not built or expanded for the needs of geotourism, but for the function of sacred tourism (Tumane Monastery). The site is in the vicinity of the tourist attraction of regional importance in which rural tourism, cultural tourism, events, excursion tourism, etc. are also developed. The position of G1 and G2 might be enhanced by better cooperation with tourist organizations, enhanced promotion, and investments. Interpretative panels are placed next to both sites at the time of applying for the status of a global geopark. They contain the data on the dimensions, geomorphological, and hydrological characteristics of these geosites, the map with the accurate location within the geopark, etc. A high-quality tour guide service for both sites could be obtained from the management of the Djerdap National Park, while the state of infrastructure and tourism-related services in the vicinity of the sites is not sufficiently high and could be improved.

The findings are in line with previous studies [66–71] that have reported that the attitudes of the local community in different regions often differ from the opinions of experts and/or visitors in regard to the importance of examined geosites, which is the reason why it is essential to take into account the overall public opinion in geotourism development. This equally applies to the locals who live near the geosite [66,67], but also to members of local authorities and municipal governments who can legally encourage and support the development of sustainable geotourism and the affirmation of the geosite [68–71].

Furthermore, the data presented in Table 2 illustrate the scores for main values (MV) = 5.07 and additional values (AV) = 6.14 as the average values in M-GAM-1-2 model. In particular, the G2 site had the highest total of main values according to the opinions of the respondents among the local population (MV = 6.26), while G1 received the highest grades from the members of the local authorities/government (MV = 5.66).

Nevertheless, the additional values of both of sites received the highest scores from the experts (AV = 9.07 and AV = 5.59). It is interesting to point out that the G1 site has lower main values than G2 according to the opinions of the experts and local population, except for the ones of the local authorities. In contrast, for additional values, the indicators for G1 are significantly better in all groups of respondents in comparison with G2. In this way, the values undoubtedly point to the fact that G1 is significantly more developed regarding the functional and tourism values (as it has already been pronounced in this study), while for G2 it can be stated that its strengths are science, education, scenic, and aesthetic conditions, along with site protection.

Based on these results, the data presented in Figure 8 show that the positions Z(i, j) of the fields in the M-GAM-1-2 matrix changed in comparison with the M-GAM matrix. In the case of M-GAM-1, G1 has more indicators in the MV, while the AV is the same as in M-GAM. This points to the fact that the local population valued scientific, educational, aesthetic indicators as well as protection factors more in comparison with the experts, which is the result of their lower criteria and the absence of professional expertise when compared with the experts [74]. Additional values are similarly assessed, which is possible for numerous reasons: Tourism and functional values are clearly defined and no space for random interpretation exists; therefore, in this case, the situation is as expected.

On the contrary, in the case of M-GAM-2, G2 has higher main values and lower additional ones in the matrix. This can be explained by the fact that members of the local government as well as a part of the local population find the indicators within these parameters to have greater importance in their local setting. This is possibly due to the fact that they do not have many capacities of objective perception of scientifically based sources and the situation is present in other regions. The lower additional values are given for their businesses and political responsibilities, and they believe that there should be more investments in the functional and tourism facilities in the observed sites.

Generally, both sites, G1 and G2, slightly moved to different positions as indicated in Figure 8. This is mostly due to the decrease in AV and modification of the position of the MV. Moreover, this could be justified by the higher level of objectivity and rigor by experts, which had an impact on the disadvantaged position of both geosites in their evaluation. Regarding the MV, the results stressed the equal importance for each group of respondents; therefore, they did not significantly affect the position of the sites.

## 6. Conclusions

The study pointed out a quantitative evaluation of the two representative geosites in the Djerdap UNESCO Global Geopark—the first global geopark in Serbia established in 2020. The tufa deposits near Tumane Monastery (G1) and Beli izvorac tufa deposits (G2) were the selected sites for consideration in the paper due to their outstanding scientific, educational, tourism, and public significance in Serbia and the Carpathian region; however, to date, they were still insufficiently investigated in academia. To achieve the assessment, the study introduced the upgraded M-GAM model (modified geosite assessment model) involving the examination of not only the opinions of visitors and experts in the field, but also the views of the local people (M-GAM-1) and local authorities (M-GAM-2) in the area. The integrated perception of the significance and role of the selected geosites would be created and sustained in science, the society, the local community, and the economy. The main objective was achieved by investigating the applicability of each suggested subindicator by submitting the importance factor in the newly introduced model. In this regard, the mission of the paper was directed toward the assessment of the significance of the observed sites in light of geotourism development.

The research findings revealed that the scores for main values are 5.07 and for the additional values are 6.14, as the average values in M-GAM-1-2 model. This indicates that the G2 site has a higher sum of the main values by the local population (MV = 6.26), while G1 has the highest grades by the local authorities (MV = 5.66). On the other hand, the additional values of G1 and G2 both received the highest scores from the experts in the field (AV = 9.07 and AV = 5.59), while the main values for the G1 site are lower than the G2 site. On the contrary, the indicators for G1 are higher in all the groups of respondents in comparison with G2 for additional values. According to these results, it can be stated that the tufa deposits near Tumane Monastery (G1 site) are more developed regarding the functional and tourism values (as it has already been pronounced in this study), while the Beli izvorac tufa deposits (G2 site) have a better position in science, education, scenic, and aesthetic perspectives, as well as the higher protection level. It is important to note that by applying this research as M-GAM-1-2, the previous M-GAM model could be considered as significantly enhanced by adding two important target groups (local people and authorities). On the other hand, it turned out that to determine the real situation of the geosites, the experts' and visitors' opinions are essential, since the local community members often do not have an objective insight into all the values of the geosites. In this respect, a comprehensive model for the assessment of the observed sites was proposed for the evaluation of the present condition and guidelines for other geosites in Serbia and abroad.

The main limitation of the study is the strong monitoring approach of the analysis rather than an explorational geosite identifying research. Therefore, it can be concluded that the research needs to be upgraded with a more reliable interdisciplinary approach taking

numerous fields into account (geologist analyses, geographical research, karst knowledge, managerial expertise, etc.) in order to achieve a more complex frame of the research. This would shed more light on the topic and reinforce future studies related to geosites and geoparks development. Moreover, this study revealed that the perception of the selected geosites differs at multiple grades and that there are no other sites that would stimulate the comparison and more comprehensive overview. In addition, it is very difficult to achieve precise answers regarding all subindicators, and the small number of answers could not be suitable for determining more objective results.

As a main contribution of this paper, the authors introduced new stakeholders in the study (local people and authorities). This upgrade revealed that the overall score can be significantly changed and the results may refer to wider groups and have a higher significance. This encourages the suggestions for future development which should be centered upon the improvement of the management system of the selected geosites, especially on the relation between national park–geopark–local authorities–tourism organizations of the municipalities. This would enable even more efficient preservation and maintenance of natural values, their organization and tourism interpretation, further research, monitoring, conservation, damage recovery, landscape arrangement, etc. Additionally, this can be achieved more efficiently by including the local population in the activities related to the protection, organization, and interpretation of tufa deposits as important sites of geoheritage, as well as the compensation for the damage and losses that occurred using these sites primarily for tourism purposes. Better tourism accessibility of both sites in the geopark, as well as their better integration with the surroundings, will be provided by the maintenance of the international waterway and the modernization and construction of the accompanying infrastructure, modernization of roads, development of public transport, arrangement of the European Cycle Path Number 6, and the European Footpath Number 4. Furthermore, an important consideration is the connection of the municipalities within the newly established global geopark, as well as the international and cross-border cooperation with neighboring Romania and other countries of the lower Danube basin. Gaining economic profit based on the establishment of special purposes of the geopark area will certainly have a stimulating role for the local community, which is especially valuable in less advantaged rural areas. The implications of the study can be reflected on a greater significance for the preservation of tufa deposits as important sites of geoheritage within Djerdap UNESCO Global Geopark. This lies in the prohibition of activities and works that could negatively affect, disturb (damage), or destroy the registered and newly discovered values of geoheritage. In addition, this indicates that natural monuments should be in the function of education and presentation of the sites of geoheritage, under the condition of controlled tourists' visits, and with the aim of the best possible preservation of natural assets.

**Author Contributions:** Conceptualization, M.D.P. and D.L.; methodology, M.D.P. and D.L.; software, M.D.P. and M.M.R.; validation, M.M.R., Đ.M. and S.K.; formal analysis, M.D.P. and D.L.; investigation, M.D.P. and D.L.; resources, T.G., Đ.M. and I.B.; data curation, T.G., S.K. and I.B.; writing—original draft preparation, D.L.; writing—review and editing, M.D.P.; visualization, D.D.B. and J.A.S.; supervision, M.K.; project administration, M.K., D.D.B. and J.A.S. All authors have read and agreed to the published version of the manuscript.

**Funding:** This research received no external funding.

**Data Availability Statement:** Not applicable.

**Acknowledgments:** The authors gratefully acknowledge the assistance received by the journal's editorial team and four anonymous reviewers. Their comments were very much helpful and appreciated. In addition, the authors would like to thank Jelena Ivanović Madžoski for supporting the research and helping with the figures adaptation.

**Conflicts of Interest:** The authors declare no conflict of interest.

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
