# Peer review of "How Can Tufa Deposits Contribute to the Geotourism Offer? The Outcomes from the First UNESCO Global Geopark in Serbia"

_land, doi:10.3390/land12020285_

Round 1

Reviewer 1 Report

Dear Author(s),
Thank you for the opportunity to read the paper entitled How can tufa deposits contribute to the geotourism offer? The 2 outcomes from the first UNESCO Global Geopark in Serbia.

I found this paper interesting. The topic of this paper is interesting but some improvements would be appreciated.

Introduction

Comment 1

Lines 48-54 – Three sentences only two references, in the end. Literature is wide when it comes to geotourism. Citing some more relevant papers would be appreciated.

Also “the majority of them use qualitative analysis ” you should cite them.

Comment 2

Line 75 – “Mission”. Avoid using “mission” in these kind of articles. “goal” “aim” objective”…

Comment 3

At the end of the section, you can mention how sections in the manuscript have been organized.

Comment 4

·       Segment Literature Review is missing.

This article should also have a theoretical background on geotourism, geosites, GAM model, etc. Point out the previous research that used GAM in Serbia and in the world. This could be used to compare your results with previous research in the discussion section.

2. Study Area and Overview of the Selected Sites

Comment 5

Lines 114-125 only one reference. Geosites and geoheritage of Djerdap are really popular in the literature.

Comment 6

In lines 131-134 you should cite each of these categories. For example Ramsar sites (Ramsar, year), IBA (Important Bird Area) (BirdLife International, year), IPA (Important Plant Area) (Plant Life year).

You should also check if is Djerdap part of the water-related PAs in the Danube basin (ICPDR, year), and the Danube Network of PAs (DanubeParks, year).

Comment 7

It would be nice to have a map showing the position of Djerdap in Europe.

3. Materials and Methods

Comment 8

The quality of Figures 6 and 7 should be improved.

Comment 9

Lines 299-312 It is not understandable from where you adapted the questions for the research. You should clarify that.

Comment 10

What was the sampling technique?

4. Results

Comment 11

The tables look very nice and easy to understand. Good job author/s!

5. Discussion

Comment 12

It would be nice to compare your results with previous studies (cases from the world, close European countries and Serbia).

6. Conclusion

Comment 13

It is not needed to repeat the result numbers in the conclusion. Focus on showing the originality of your work. Novelty? What are theoretical contributions? Management implications?  What are the limitations? These segments are missing.

Once again thank you very much for the opportunity to read this interesting article. The manuscript has really nice results, but improvements would be appreciated. Looking forward to reading your article again.

Wish you all the best!

Sincerely,

Reviewer 

Author Response

Dear Reviewer,
Thank you for your kind words and encouragement. We do appreciate your useful suggestions and tried to implement most of them. Please, find the attached document with our comments point by point.
Sincerely,
Authors of the paper

Reviewer 2 Report

Manuscript: How can tufa deposits contribute to the geotourism offer? The 2 outcomes from the first UNESCO Global Geopark in Serbia.

The manuscript presents an interesting way of evaluating geosites. To this end, the methodology based on the set of indicators is shown in some detail. Two tufa deposits sites located in the Djerdap UNESCO Global Geopark in south-eastern Serbia were selected for the assessment.

The article is well planned and has many valuable features. Especially the use of the opinion of residents and tourists visiting these places in my opinion is a great idea. As a geologist working in the field, I feel that such a site should have a slightly more extensive geological description, but there is no geologist among the authors.

The biggest drawback of the work is the graphic side. This mainly applies to Figs. 2 and 3. Notes below. Figures 4 and 5 without descriptions do not make any more sense to a potential reader than colored slacks. Especially since the photos are composed of small windows. I suggest giving twice as large photos divided into two sections.

I have marked the errors that could not be avoided and significantly reduce the quality of the entire article. They must be removed:

Line 33: abbreviation “M-GAM-1-2” cannot be a keyword

Line 40: two very general sentences are affiliated with 8 citations. Of these, 4 items are works by Hose T. With all due respect to this author, this is too much.

Line 52-54: I am clearly missing information that sustainable economic development is an important goal

eg. ROSADO-GONZÁLEZ E.M., SÁ A.A., PALACIO-PRIETO J.L. 2020 –UNESCO Global Geoparks in Latin America and the Caribbean, and Their Contribution to Agenda 2030 Sustainable Development Goals. Geoheritage, 12 (36); https://doi.org/10.1007/s12371-020-00459-2

Line 78: explain the abbreviation used for the first time in the text

Line 103: not World this is Global Geoparks Network

Figure 116. Added citation of Fig.2 as Fig.1

Figure 1. and 2 change the order

Line 137: I am a geologist and I admit that I rarely come across such rocks. Therefore, I propose to add the information that tufa is synonymous with calcareous sinter. Or maybe it should be added that it is a type of travertine? Otherwise, someone might think that it is about volcanic rocks: tuffs

Cite: A.Pentacost, 2005, Travertine. Springer

Figure 2. I proposed Fig. 1, geographical names are illegible, too small, You must insert local names used in the text: Danube River, Kamenica, Tilva Hill, Krś Mountain etc.

Figure 3. These figures are not printable, they are field sketches. In addition, after the symbols G1 and G2, they have been stretched without maintaining the scale, which completely degrades their content and credibility. The symbols are non-standard, incomprehensible without the legend.

Line 154: Did the authors recognize that these were Cretaceous limestones? If not, please cite the geological map of the area or other work on the geological structure of the research area.

Figure 4. photos without descriptions are completely useless. If the authors have no information for the reader what it is, I suggest removing it. However, if there is something interesting there and they describe it, I suggest you split it into A, B, C and add the names of the authors

Figure 5. photos without descriptions are completely useless. If the authors have no information for the reader what it is, I suggest removing it. However, if there is something interesting there and they describe it, I suggest you split it into A, B, C and add the names of the authors

Figure 6. increase by 15%

Line 264: cite who made the declaration

Figure 7. increase by 15%

Line 351: which G1 or G2

Author Response

(The authors gave the same response as above.)

Reviewer 3 Report

Dear Authors,

Please find my few comments here and a few technical corrections in the attached doc file.

Abstract and Intro: Very good! I am eager to readt the paper! :)

Study area: it is good in overall.
- Fig. 3 a/b.: The figures should not be distorted. In this way, it is more difficult to read and the scale is useless.
- Nice photos!

Materials and methods: nice!
- Figs 6 and 7.: the font size is too small, the figures can be hardly read.

Results: nice!
- Tbale 1.: I would change the column order of complete scores and scores by the different groups.

Discussion: it is a very long and detailed discussion chapter. Please consider mentioning the not statistical amount of questionnaire fills. I also did some research using M-GAM, and it is very hard to get answers regarding all 27 subindicators. And the small number of fills is unfortunately not suitable for determining objective results. Imagine if 1 or 2 local government member is replaced from the 11, it could give quite different results... You should emphasise somewhere in the beginning that this kind of analysis is not for an explorational geosite identifying research, but for monitoring instead.

Conclusions: OK!

Overall: I am very happy to read this article, and it gave me inspirations for my future work, I will consider trying out M-GAM 1 and 2. My minor revision decision is only because of the small corrections that I asked for above.

Best regards,

Author Response

(The authors gave the same response as above.)

Reviewer 4 Report

Dear authors,

Thank you for the opportunity to read your paper. The study focuses on the present state and the assessments of geotourism development of the two most representative tufa deposits in the Djerdap National Park – the first UNESCO Global 21 Geopark in Serbia. Tufa deposit forms are very picturesque; therefore, the choice of the topic is very good. The Introduction (Literature review) must be widened. The structure of the paper is clear. The methodology is correct. Results are valuables. Discussion is concise. Conclusions are forward-looking. However, the References include only 53 items. This must be widened. More "sustainability" papers are needed.

Author Response

(The authors gave the same response as above.)

Round 2

Reviewer 1 Report

Thank you for replying to my comments. I believe the article is very much improved now and it can be accepted for the publication. 

Wish you all the best!